# Meaningful Engagement of Persons Affected by Leprosy in Research: An Exploration of Its Interpretation, Barriers, and Opportunities

**DOI:** 10.3390/tropicalmed8010052

**Published:** 2023-01-10

**Authors:** Laura de Groot, Anna T. van ‘t Noordende, Mathias Duck, Joshua Oraga, Sarju Sing Rai, Ruth M. H. Peters, Nienke Veldhuijzen

**Affiliations:** 1Leprosy Research Initiative, 1097 DN Amsterdam, The Netherlands; 2Athena Institute, VU University Amsterdam, 1081 HV Amsterdam, The Netherlands; 3NLR, 1097 DN Amsterdam, The Netherlands; 4International Federation of Anti-Leprosy Associations (ILEP), 1204 Geneva, Switzerland; 5The Leprosy Mission International, Brentford TW8 0QH, UK; 6International Association for Integration, Dignity and Economic Advancement (IDEA) Kenya, Nairobi 00200, Kenya

**Keywords:** leprosy, public and patient involvement, meaningful engagement, experts-by-experience

## Abstract

Despite the growing interest in public and patient involvement in research, best practices in the leprosy context have yet to be explored. This mixed-method study aimed to explore the interpretation, barriers and opportunities of meaningful engagement of persons affected by leprosy in research through: (i) an exploratory phase consisting of key informant interviews with experts in public and patient involvement (*n* = 2) and experts-by-experience (i.e., persons affected by leprosy; *n* = 4), and (ii) an in-depth phase among leprosy researchers consisting of an online survey (*n* = 21) and key informant interviews (*n* = 7). Qualitative data were thematically analyzed. Basic descriptive statistics were used to summarize the survey data. Key informant interviewees unanimously agreed to the importance of engagement in research. Survey results indicated that the level of engagement differed across research stages. Identified barriers included a lack of skills for or awareness of engagement among both experts-by-experience and researchers, stigma and limited time and resources. Opportunities included capacity strengthening, creating a shared understanding, building rapport, and establishing a safe environment. In conclusion, this exploratory study emphasized the importance of engagement of experts-by-experience in leprosy research and identified ways forward that include, but are not limited to, the acknowledgement of its value and creating a shared understanding.

## 1. Introduction

Leprosy, also known as Hansen’s disease, is a neglected tropical disease caused by the pathogen *Mycobacterium leprae*, which mainly affects the peripheral nerves [1]. This often leads to nerve function impairment, which may result in permanent physical impairments and disability. Although the introduction of multi-drug therapy resulted in a decline in the prevalence of leprosy, new cases continue to occur and for the past 15 years approximately 200,000 new patients are diagnosed annually [2]. Most new leprosy patients are found in impoverished communities in low- and middle-income countries [3]. Although the disease is curable, many persons affected by leprosy continue to experience social stigma and discrimination [4,5,6,7].

In 2019, the “Leprosy Research Initiative” (LRI), a research funding and capacity strengthening organization, conducted a stakeholder consultation to investigate research priorities in the field of leprosy. This study revealed a wish to enhance participation of persons affected by leprosy in research, which would also help to reduce stigma and discrimination [8]. This growing demand for participation is in line with current trends in leprosy services in general [9] and is also perceived in the wider field of health research, across many other diseases and health conditions worldwide [10,11,12].

Particularly in high-income countries, developing stronger public and patient involvement (PPI) in research has gained momentum since the beginning of the 21st century [13,14,15]. Perceived benefits include a deeper understanding of patient issues, increased research quality, and diminishment of the gap between science and practice [16,17]. Due to the wide variety in terminology used to describe PPI practices and generally poor documentation in research, insights into the extent and impact of such practices in low- and middle-income countries remain limited. However, a review of Cook et al. (2019) demonstrated that PPI across different research stages and engagement levels does take place in low- and middle income countries [18]. This review showed that regardless of the subject, type or setting of research, PPI can be integrated in the research process. Even though PPI is often not used in all stages of the research cycle, evidence suggests that patients can have meaningful contributions in all stages [19].

Despite the growing demand for and desire to involve persons affected by leprosy in research, best practices for achieving meaningful engagement have yet to be explored. Therefore, this study aimed to explore the interpretation of meaningful engagement of persons affected by leprosy in research from the perspectives of experts-by-experience (i.e., persons affected by leprosy), leprosy researchers and experts in PPI, and to understand the barriers and opportunities in the leprosy context. To address this aim, we used a mixed methods approach in which the qualitative component allowed us to explore in-depth the interpretation of and experience with meaningful engagement while the quantitative component provided insight in the frequency of engagement practices in the leprosy context.

### The Concept of Patient Engagement

As the concept of public and patient involvement becomes increasingly accepted and its value more widely recognized, so has the number of terms and definitions used to capture the same phenomenon [20]. Although boundaries between terminology to capture the phenomenon of engagement remain unclear sometimes, the terms ‘involvement’, ‘participation’, ‘activation’ and ‘empowerment’ are mainly targeting the patient, whereas ‘engagement’ also focuses on the partnership between patient and researcher [21]. The present study considers perspectives of both leprosy researchers and persons affected by leprosy, and therefore refers to ‘engagement’. A further nuance to the use of ‘patient’ is that within leprosy service delivery and research, people no longer receiving treatment are not referred to as patients, but as persons affected by leprosy [9]. Therefore, in the context of this paper, we define ‘patient engagement’ in research as “*the practice of collectively co-building research programs through meaningful and equal partnerships between persons affected by leprosy and scientists*” [19], and ‘patients’ *as “individual[s] with personal experience of leprosy*” [22]. In recognition of their expertise on account of their personal experience and in line with the terminology used in the wider field of engagement, we will refer to persons affected by leprosy as “experts-by-experience” in this study. Furthermore, we distinguish three levels of engagement: (1) consultation (i.e., opinions and views of experts-by-experience are asked and inform the decision-making process), (2) collaboration (i.e., effective partnerships are built between researcher and experts-by-experience, so shared decision-making is driving the research process) and (3) initiated/led by (i.e., the research process is controlled, directed and managed by experts-by-experience, in which researchers have a supportive role) [23].

## 2. Materials and Methods

### 2.1. Study Design

For this study, a cross-sectional design with a mixed-methods approach was used including a qualitative component with key informant interviews and a quantitative component with an online survey. In the exploratory phase (a), key informant interviews with experts in PPI, and with experts-by-experience were conducted to explore their interpretation of meaningful engagement and to identify barriers and opportunities in the leprosy context. These exploratory interviews were used to inform the in-depth phase (b, c), consisting of a survey among LRI-funded researchers and key informant interviews with leprosy researchers (Figure 1). The aim of the survey was to investigate current levels of engagement throughout the research cycle and researchers’ experiences in this, in terms of experienced barriers and perceived outcomes and impact. The aim of the key informant interviews with leprosy researchers was to discuss the findings from the exploratory phase more in-depth, and to explore potential opportunities to enhance engagement in the leprosy context.

### 2.2. Participant Engagement

Two expert advisors with personal experience of leprosy (M.D. and J.O.) were involved throughout the project to co-develop the research proposal, review the data collection instruments (i.e., interview guides and survey design), and to help interpret the results.

### 2.3. Study Population and Sample Size

Three groups of participants were included in our study: (1) experts in PPI, (2) experts-by-experience, and (3) leprosy researchers. Experts in PPI were eligible if they had working experience in both the disability and the leprosy field. Five of the seven leprosy researchers included in the interviews had no current LRI funding nor were lead applicants in the 2021 budget round. Researchers were eligible to participate in the survey if they were lead applicants of a research project that received funding from the LRI between 2015 and 2020. Individuals were excluded if they were unable or unwilling to give informed consent and/or unable to speak English. Interviews were conducted until data saturation was reached. The number of participants in the survey was maximized by the total number of researchers receiving LRI funding during the study period.

### 2.4. Sampling Strategy

Contact details of potential participants were obtained via the network of the LRI and one of its partner organizations NLR (until No Leprosy Remains). Subsequently, these potential participants were approached by email. For the interviews within the exploratory phase with experts in PPI and experts-by-experience, purposive sampling was used to recruit a diverse sample [24], in terms of global regions. For the interviews with leprosy researchers in the in-depth phase, purposive sampling was used to include researchers with different academic backgrounds (i.e., social sciences or public health, laboratory sciences, health systems research, translational research, clinical research, epidemiology, or operational research). For the survey, a list was created of the lead applicants of all research projects funded by the LRI between 2015 and 2020. Contact details were obtained via the LRI-database and invitations were sent out by email.

### 2.5. Data Collection

Data were collected from April to June 2020. The key informant interviews were semi-structured. The interview guides were developed based on literature and adapted for the different stakeholder groups—experts in PPI and experts-by-experience. The interview guides covered the following themes: (1) interpretation and importance of meaningful engagement, (2) process of engagement, (3) engagement across research stages, (4) outcomes and impact, (5) barriers, and (6) enablers. Based on the preliminary findings of the exploratory phase, the interview guide for leprosy researchers in the in-depth phase was developed to explore these topics more thoroughly. To gain an understanding of the nuances of engagement, some topics (e.g., the meaning of meaningful engagement) were not restricted solely to the exploratory phase, as these were explored with all stakeholder groups. To pre-test the sequence and interpretations of topics and questions, a pilot interview was performed for all three interview guides. All key informant interviews were conducted online by the same researcher (L.d.G.), using the meeting platform Zoom.us and lasted approximately one hour. After each interview, the interviewee received a summary of the interview, to check if it correctly reflected what was discussed. 

The online survey consisted of eight sections: (1) general information, (2) information about research experience, (3) levels of engagement and research stages, (4) objectives and means of engagement, (5) levels of engagement achieved/expected to achieve, (6) challenges, (7) outcomes and impact and (8) engagement by other stakeholder groups. To capture the stages of the research process, the seven stages described by the National Institute for Health Research (NIHR) were used (i.e., identifying and prioritizing, commissioning, designing and managing, undertaking, disseminating, implementing, and evaluating impact) [25]. Prior to the dissemination of the survey, a pilot test among two people was carried out to pre-test the sequence and interpretations of questions and slight adaptations were made. The survey was created using Google Forms. Two weeks after the initial invitation, a reminder was sent. Participants had three weeks to complete the survey.

### 2.6. Data Analysis

Audio-recordings of all interviews were transcribed verbatim. For qualitative data analysis, thematic analysis was performed using the software program Atlas.ti [24]. After coding the first three transcripts, a codebook was constructed. New themes that emerged while coding the remaining transcripts were added to the codebook. Quantitative data analysis was performed by using the SPSS Statistics version 24.0 (IBM Corp, Armonk, NY, United States). Basic descriptive statistics were used, such as means for continuous variables and proportions for categorical variables. Open-ended questions in the survey were analyzed by performing thematic analysis [24]. Data were analyzed by two independent researchers (L.d.G. and A.T.v.‘t.N).

## 3. Results

In the exploratory phase, six participants were included. Two participants were experts in PPI and four participants were experts-by-experience. None of participants in the exploratory phase were recipients of funding from the LRI. For the in-depth phase, 44 invitations were sent to researchers, 21 agreed to participate (response rate 48%). We checked whether the origin of the LRI-funded researchers who refused to participate in the study were systematically different from those who participated and found this was not the case. In addition, seven leprosy researchers participated in the key informant interviews. Two of the seven leprosy researchers included in the interviews received funding from the LRI at the time of the interview; five had received funding in the past. A few participants taking part in the key informant interviews were initially invited to participate from a specific participant position but were also able to share experiences from another position (e.g., an expert in PPI who was also expert-by-experience).

Most participants were from leprosy-endemic countries (*n* = 27, 79%; *n* = 5 in the exploratory phase and *n* = 22 in the in-depth phase). Of the researchers who participated in the survey, 13 researchers (62%) had received funding from the LRI before 2018, and eight researchers (38%) had received funding in 2018 or later. From 2018 onwards, the LRI specifically asks researchers to indicate how persons affected by leprosy will be engaged in their research projects. Participant characteristics are summarized in Table 1.

### 3.1. Meaningful Engagement

While all participants were asked what they consider meaningful engagement, only a few participants were able to express what they consider it to be. Engagement was perceived as meaningful when (i) experts-by-experience have the knowledge and skills to contribute, (ii) when the process reflects equal relations between researchers and experts-by-experience, and subsequently, (iii) when the potential outcomes are perceived as beneficial and impactful for all parties involved. Participants emphasized that engagement would result in research outcomes and impact that are much more meaningful to persons affected by leprosy.

### 3.2. Importance and Relevance of Engagement

According to experts in PPI, the starting point of engagement is acknowledging the value of the perspective of experts-by-experience that could be brought into research by meaningfully engaging them:


*“I think that for any problem, anything in life, we need different point of views”*
(Expert in PPI, but also expert-by-experience, Asia)

All participants in the interviews, despite their differences in background, origin, and stakeholder position, unanimously recognized the importance of engagement in research. Participants explained that engagement is important because experts-by-experience bring in a different perspective than researchers, have the best understanding of their issues and needs, and their involvement leads to participants ‘opening up’ more.

As one participant explained:


*“They know the right questions to ask, they know how to ask them. They have the right language, and people, their peers, that they would interview [and] would open up far more readily than [to] a complete outsider”*
(Expert in PPI, Oceania)

Another participant explained:


*“This is very important. Why? He who wears the shoes knows where it pinches”*
(Expert-by-experience, Africa)

As part of the survey, participants were asked to indicate to what extent they agreed with six statements regarding the relevance, quality, and learnings for both researchers and experts-by-experience of the engagement of experts-by-experience in their research. Findings of these statements are presented below in Figure 2.

Figure 2 shows that most researchers felt that the engagement of experts-by-experience contributed to the relevance of their study—76% (strongly) agreed. This is aligned with the qualitative finding that some leprosy researchers mentioned that engaging experts-by-experience increases the societal relevance of their work.

Furthermore, findings of the survey showed that 67% (*n* = 14) of the researchers agreed or strongly agreed that by engaging experts-by-experience in their work, they have learned about the knowledge and perspectives of experts-by-experience. Additionally, most researchers (*n* = 14, 67%) agreed or strongly agreed that engagement has directly influenced experts-by-experience, in terms of knowledge, skills and societal support. From the perspective of experts-by-experience, qualitative findings showed that potential learning experiences and capacity strengthening deriving from engagement are important outcomes for them. Most experts-by-experience felt that being engaged in research could be a powerful way to empower them in their skill development and attitude towards their personal experience of living with leprosy. Additionally, some experts-by-experience described that being engaged in research would also be an opportunity for them to demonstrate their capacity.

One participant explained:


*“Research contributes to show the capacity of the persons affected in different aspects of daily life and society, not only as being observed, but how to transform reality and produce knowledge”*
(Expert-by-experience, Latin America)

Another participant told us:


*“Being engaged in a research project could be a life changing event”*
(Expert-by-experience, Asia)

### 3.3. Engagement across Research Areas

Although leprosy researchers from different research areas valued the importance of engagement in their work, they agreed that the level of engagement that could be applied depends on the specific research area and type of study. Researchers felt that study objectives should reflect and recognize the additional value of engagement. Both experts in PPI and some leprosy researchers explained that in fundamental research projects engaging experts-by-experience is often not feasible.

One participant explained:


*“The patients that are involved in my projects don’t ask me for feedback because my research is so basic that it is impossible [for them] to give feedback”*
(Leprosy researcher, Brazil)

Some participants described that epidemiologists are mostly interested in the causes and effects of leprosy, whereas social scientists primarily focus on the impact of leprosy on human lives—therefore, they felt that qualitative methods are much more aligned with the idea of engagement compared to quantitative methods.

Some participants explained that engagement should be a bottom-up process, created collectively with experts-by-experience:


*“It is like democracy, you can’t pose democracy; democracy has to be born from within”*
(Expert in PPI, Oceania)

### 3.4. Levels of Engagement across Research Stages

When asked in the survey why researchers would engage or had engaged experts-by-experience across the various research stages of their research projects, improved reflection of perspectives of experts-by-experience and improving knowledge translation were mentioned most often. Increased chance of funding was mentioned only twice, and empowerment of persons affected by leprosy was also not mentioned often. According to the participants in the survey, experts-by-experience were engaged most frequently in the ‘design and management’ and ‘undertaking’ stages of research, and most infrequently in the ‘identifying and prioritizing’ and ‘evaluating impact’ stages.

Resulting from the survey, Figure 3 presents the levels of engagement that are expected to be achieved or that have been achieved—depending on whether the project was completed yet or not—indicated per research stage.

During the first stage of identifying and prioritizing, 33% (*n* = 7) of the researchers did not engage experts-by-experience or solely engaged them for information, and 38% (*n* = 8) did engage them in terms of consultation—the levels of collaboration (*n* = 5, 24%) and initiated/led by (*n* = 1, 5%) remained low. Looking at further stages of the research process, the number of researchers that did not engage or for information only is decreasing, whereas the level of collaboration is increasing over stages. Overall, findings showed that the level of initiated/led by was barely achieved in any of the stages—the stage of undertaking research peaked in this (*n* = 4, 19%). When looking at the average level of engagement across research stages of projects before and after 2018, there is no indication that engagement had increased in the latter group.

### 3.5. Barriers to Engagement in the Leprosy Context

Many participants in the interviews mentioned that not every person affected by leprosy would be able to meaningfully engage in research. Participants felt that this was mostly related to a lack of education and illiteracy, and a lack of certain skills (e.g., research, writing, and communication skills) of experts-by-experience. One participant explained:


*“When you are applying for funds in the international context, you have to speak very good English, you have to write very good research questions, the methods should be very precise and clear. The project expectations and plan should be met, and that is very challenging to [uneducated] persons affected by leprosy”*
(Leprosy researcher, social sciences, Asia)

Another participant said:


*“You have to have some credentials before you are being assigned to any research project and most of persons affected by Hansen’s disease [leprosy] lack education”*
(Expert-by-experience, Africa)

There were also many participants who considered stigma a barrier to engagement. Community stigma and the issue of concealment were mentioned most often, but internalised stigma and a lack of self-confidence were also considered barriers:


*“They do not want to come openly since the stigma, the discrimination still exists. These are the reasons I think, that people affected are not involved in research”*
(Expert-by-experience, Asia)

In addition, many participants mentioned that meaningful engagement is a time-consuming process, whereby it is aimed that direct outcomes will persist in the later future. A few participants mentioned that in the field of leprosy, engagement and participatory approaches are not valued or understood. In addition, some participants mentioned that there is a lack of funding for leprosy in general, and especially for participatory research. Finally, embracing the perspective of an ‘outsider’ was often perceived as difficult:


*“And then all of a sudden there is another person, who is also experienced, but has a different perspective”*
(Leprosy researcher, social sciences, but also expert-by-experience, Latin America)

As part of the survey, researchers (*n* = 21) were asked to indicate the barriers experienced across the different research stages. Barriers which were experienced by the majority of researchers related to the lack of knowledge about engagement strategies by researchers (*n* = 13) and to difficulties in recruiting experts-by-experience (*n* = 14). The former was mentioned across all research stages, while the latter was mostly experienced during the first three research stages (i.e., identifying and prioritizing, designing and managing, and undertaking). Six researchers indicated the barrier related to insufficient research skills of experts-by-experience, mostly at the stage of undertaking. Practical challenges such as financial constraints (*n* = 8) and time consuming (*n* = 7) were found throughout the entire research process but least frequently at the stage of identifying and prioritizing.

### 3.6. Opportunities for Enhancing Engagement in the Leprosy Context

Several opportunities were mentioned by the participants in the interviews. These were often related to the barriers. For example, many participants mentioned that an important role of the research team is to provide training to experts-by-experience in order to build research capacity, which is needed to genuinely contribute to research. While this was considered an essential part of the engagement process, it was also found to be an opportunity. One participant said:


*“It is good to enhance that capacity, their capacity, so that they can join with us”*
(Expert-by-experience, Asia)

Creating a shared understanding, building rapport between researchers and experts-by-experience, and establishing a safe environment for engagement were considered important opportunities by the participants. Participants explained that bringing an expert-by-experience on board of the research team, could also enable recruitment of other suitable persons. Some participants mentioned that organizations of persons affected by leprosy can be instrumental in identifying suitable people, who can engage in research.

Participants also mentioned that it is important that people of influence, such as government officials, community leaders, and principal investigators, are supportive of public and patient involvement in research. Trust among stakeholders was considered beneficial to move forward and promote engagement in the field of leprosy. Some participants suggested creating national policies or guidelines that encourage engagement of experts-by-experience in research. One participant told us:


*“You want to reach people with influence, and I think that is really important. Well, I think I am little, not cynical, but I am a little bit “Oh, we need to let their voices be heard.” I think voices are being heard, but they have to be heard by the right people. And that is what is really important, they have to [be] heard by people that can implement changes that will be possible”*
(Expert in PPI, Oceania)

Participants also emphasized that strategies should be tailored to the local context to be effective in order to attain the impact it aims for, and that effective communication strategies are crucial for engagement. A leprosy researcher said:


*“It depends on the context, how do local people see, feel, understand and tackle or respond to the leprosy affected persons, this is the context. And that can vary from one society to another society. Not only that, it changes over the period of time”*
(Leprosy researcher, social sciences, Asia)

## 4. Discussion

With this exploratory study, we provided insights into the interpretation of meaningful engagement of persons affected by leprosy in research from the perspectives of experts-by-experience, leprosy researchers and experts in public and patient involvement, and the barriers and opportunities for enhancement in the leprosy context. Even though the importance of engagement in research was unanimously recognized among our study participants, we observed in the interviews that only a few participants were able to express what they consider meaningful engagement. This may be explained by the wide variety in terminology being used in both literature and practice to describe engagement practices [20,21], leaving room for interpretation among different stakeholder groups. As a potential consequence, Gallivan et al. (2012) argued that this room for interpretation may lead to misunderstanding and misinterpretation of expectations, goals and outcomes of engagement by different stakeholder groups [20], posing a potential barrier to achieving meaningful engagement. Indeed, recognition of the value of engagement and consensus on its interpretation are essential starting points for a successful engagement process. Gray-Burrows et al. (2018) emphasize that clarification of the role of PPI in research is essential to optimize the use of time and efforts researchers and persons being engaged allocate to it [26]. While there is growing support for the philosophy that experts-by-experience should be “involved” and “take part” in research, also in the present study, there is no clear consensus on what this precisely means and how much, or to what extent, experts-by-experience should be included [20,27]. The lack of standardisation in designing and evaluating PPI frameworks and strategies complicates the development of a comprehensive PPI strategy in researchers’ work [27,28,29]. Over the last decade, researchers have been developing PPI toolkits, but these often lack step-by-step or practical and detailed ‘how to’ guides. For example, the WHO’s guidelines for strengthening participation of persons affected by leprosy in leprosy services include several strategies and goals to involve experts-by-experience in research, but do not provide any guidance on how to realize these in practice [9]. Although future research should further explore this need, we expect that the development of a practical and evidence-based PPI guideline or toolkit would support leprosy researchers to move towards meaningful engagement in their work. Developing such a guideline or toolkit for the leprosy context should be a key area for future focus.

Furthermore, many barriers and opportunities found in this study do not seem to be unique for the leprosy context. For instance, previous studies demonstrated that barriers such as lack of capacity of experts-by-experience to engage in research, lack of knowledge on engagement strategies by researchers and practical challenges such as financial- and time constraints do occur in a wide variety of settings and contexts, and are not solely restricted to low- and middle-income settings [30,31,32]. The same applies for the opportunities that were identified in our study, such as creating a shared understanding, building rapport, and establishing a safe environment for PPI [30,31,32]. Therefore, more focus on exchanging engagement experiences across various settings and contexts and learning from each other is desirable for enhancement. A challenge to meaningful engagement in the leprosy context, and likely for other stigmatized conditions, that was identified in the present study is internalized and community stigma, which may hamper recruitment of suitable candidates to be engaged [33]. Leprosy-related stigma and discrimination have been widely documented in literature [4,5,6,7]. While this was identified as a barrier to the leprosy field in the present study, being engaged in research was also seen as a powerful tool to empower experts-by-experience. Indeed, personal empowerment is considered the opposite of internalized stigma [34]. These findings stress the importance of stigma reduction for PPI to be successful in the field of leprosy, and at the same time, highlights that meaningful engagement can contribute to stigma reduction.

While often the promising aspects of engagement and its potential impact on research are highlighted, potential hazards should be acknowledged as well. While interpreting our study results, it is important to note that all experts-by-experience who participated in the present study were actively involved in leprosy organizations, which might have influenced the extent of representativeness of the wider leprosy community. Interestingly, Boote et al. (2010) argue that the discussion about ‘representativeness’ is already problematic in itself, as the persons being engaged are only able to represent themselves. Even expecting that these persons could represent the views of others with similar life experiences places an unreasonable burden on them [35]. The issue of representativeness also applies to the two expert advisors on this project, since both are members of the ILEP Advisory Panel of women and men with personal experience of leprosy, and thus used to regularly providing feedback and expressing their views, experiences, and opinions. On the other hand, aligned with findings of the present study, certain skills such as writing and communication are needed to be engaged in a meaningful manner [36]. Several studies found that the ability to meaningfully engage is affected by education level, level of income, age, cognitive skills and cultural differences [11,37,38]. Ives et al. (2013) describe this as the *Professionalization Paradox*: experts-by-experience should obtain adequate training in order to contribute to research, but once someone becomes professionalized by getting familiar with research to be substantially engaged, their authentic ‘insider’ perspective and thereby the level of representativeness become questionable [39]. In a commentary, Staley (2013) critically responded to this assumed ‘paradox’, stating that it does not exist in many cases of PPI; training should be about supporting experts-by-experience to understand the basic principles of research, not about training them to acquire the same skills as researchers [40]. In this discussion, realizing that different types of knowledge, experiences and perspectives complement each other is key [41]. Further, researchers highlight the need for active reflexivity to explore and clarify their positionality in order to deal with such a paradoxical issue of representativeness or lack thereof [42,43]. Understanding and being reflexive of one’s social, ontological, and epistemological locations and beliefs, and how these may influence the research process is essential in avoiding or reducing bias in representativeness [43,44].

Moreover, studies investigating engagement practices often warn of the potential occurrence of tokenism, which can be described as “*[the] difference between going through the empty ritual of participation and having the real power needed to affect the outcome*” [45]. Tokenistic or symbolic involvement in research stems from failure to address unequal power relations between researchers and experts-by-experience. For some researchers, it may be difficult to change their beliefs about who is perceived as the ‘expert’ and who is perceived as the ‘recipient’ of knowledge and expertise. In addition, tokenism may be related to the fact that researchers do not always understand and embrace the contributions that PPI could bring to research [16], which was also mentioned by participants in the present study. In that sense, the movement towards meaningful engagement may be limited by the beliefs and attitudes of researchers. A failure to address these beliefs may result in tokenistic engagement. This again stresses the need for practical guidelines on methods to achieve meaningful engagement in leprosy research. Additionally, published evidence of the impact and added value of meaningful engagement could help to ensure that PPI is understood and appreciated. Funders could also play an important role in recognizing meaningful instead of tokenistic engagement and would ultimately also benefit from it, as PPI strives to result into more meaningful research outcomes and impact [46]. Accepting that payment of experts-by-experience is a legitimate use of funds is a first and crucial step, as it acknowledges the value experts-by-experience bring to research and reflects equal relations. Finally, Hahn et al. (2017) developed a checklist for grant reviewers to evaluate genuine engagement (versus tokenistic engagement) that could be used by funders [47].

Since 2018, the LRI asks researchers explicitly to indicate how persons affected by leprosy will be engaged in their research projects. We did not find an indication that engagement had increased among the researchers receiving LRI funding since 2018. In addition, although numbers in our study are small to draw firm conclusions, the level of engagement appeared to increase from stages prior to the actual research (identifying of the research question), to stages of implementing the research (design & managing, and undertaking) to the stages after the research is completed (dissemination, uptake, and impact). Findings from the key informant interviews showed that almost all participants acknowledged the importance of engagement throughout the different stages of the research cycle. This is in line with findings from a systematic review conducted by Domecq et al. (2014), emphasizing that engagement throughout the research cycle seems to be feasible even in populations with high prevalence of social inequities, such as poverty and illiteracy [28].

### Strengths and Limitations

This study has several limitations. The potential occurrence of socially desirability bias should be borne in mind while interpreting results, particularly by taking the financial relationship between the LRI as funding organization and leprosy researchers into account [48]. This limitation was to some extent mitigated by anonymizing survey responses. While we consider the mixed methods approach of this study a strength, the generalizability of our findings could be limited by the relatively small sample sizes, in both the exploratory and in-depth phases. However, we believe that this exploratory study could be seen as a steppingstone to contribute to the further encouragement of engagement practices in the leprosy context and other related fields, such as other neglected tropical diseases and engagement practices in low- and middle-income settings.

## 5. Conclusions

This exploratory study emphasized the importance of engagement of experts-by-experience in leprosy research, demonstrated its complexity and identified ways forward that include, but are not limited to, the acknowledgement of its value and creating a shared understanding. It is recommended to engage experts-by-experience throughout the entire research cycle for optimal outcomes and impact for all stakeholders involved, although the process of engagement can vary across research stages. Funders can play an important role in both stimulating engagement, for example, by accepting that payment of experts-by-experience is a legitimate use of funds, and in recognizing differences between meaningful and tokenistic engagement. The potential contributions of engagement practices to stigma reduction and empowerment, can be an interesting area for future studies. Finally, a key area for future focus is the development of a practical and evidence-based PPI guideline or toolkit tailored to the leprosy context, supporting leprosy researchers to meaningfully engage experts-by-experience in their work.

## Figures and Tables

**Figure 1 tropicalmed-08-00052-f001:**
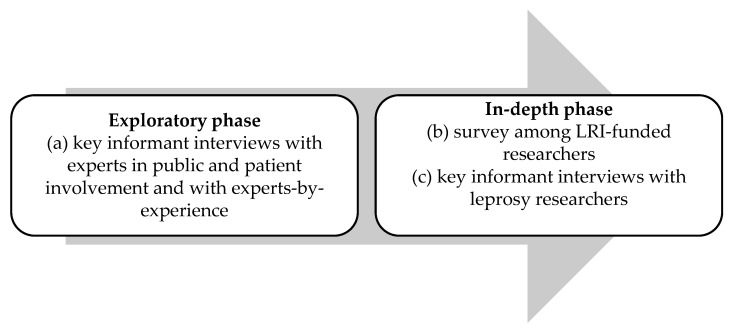
Study design.

**Figure 2 tropicalmed-08-00052-f002:**
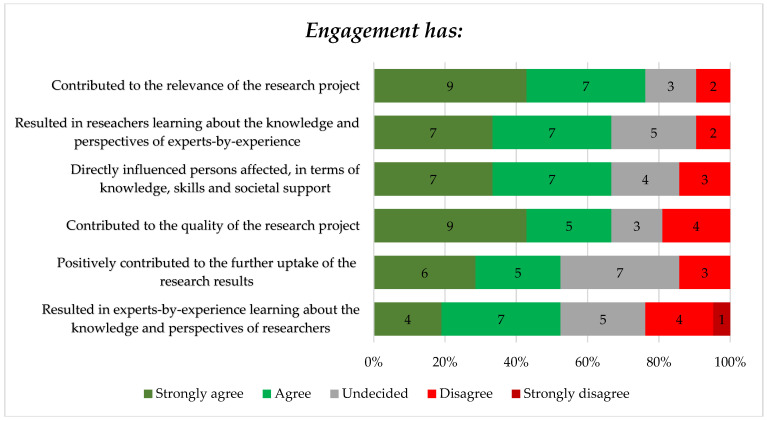
Participants’ perceived degree of agreement regarding statements about their experiences with engagement of experts-by-experience in their research.

**Figure 3 tropicalmed-08-00052-f003:**
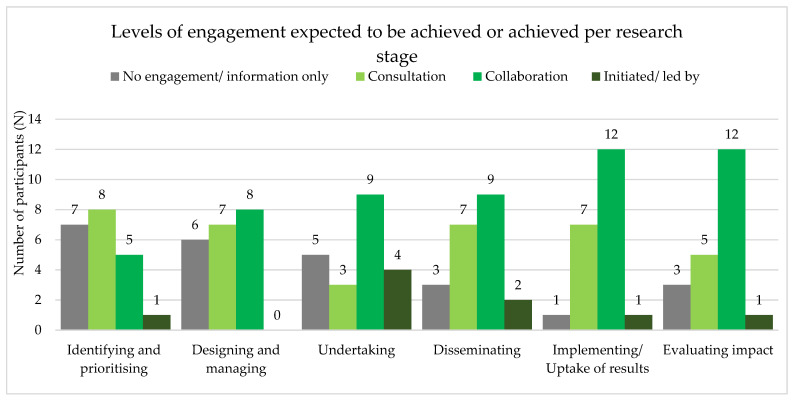
Levels of engagement achieved or expected to achieve, indicated per research stage.

**Table 1 tropicalmed-08-00052-t001:** Participant characteristics (*n* = 34).

	Exploratory Phase	In-Depth Phase	
	(a) Key Informant Interviews (*n* = 6)	(b) Survey (*n* = 21)	(c) Key Informant Interviews (*n* = 7)
Gender, *n*		N.A. *	
Female	3		3
Male	3		4
Origin, *n*			
Asia	3	10	3
Latin America	1	3	3
Africa	1	2	1
Europe	0	3	0
Oceania	1	2	0
United States	0	1	0
Experience **, *n*			
Researcher	1	21	7
Expert-by-experience (i.e., person affected by leprosy)	5	-	1
Expert in public and patient involvement in research	2	-	0
Research area ***, *n*	N.A. *		
Social sciences or public health		8	4
Laboratory/basic sciences		4	2
Health systems research		3	0
Translational/applied field research		3	1
Clinical research		2	1
Epidemiology or operational research		1	1

* Not applicable/data not collected. ** In the exploratory phase, one participant was invited as an expert in public and patient involvement, but also had personal experience with leprosy. One participant who was invited as an expert-by-experience, was also a researcher. In the in-depth phase, one leprosy researcher also had personal experience with leprosy. *** Multiple categories possible; therefore, the total number exceeds the number of participants.

## Data Availability

All data (both qualitative and quantitative) generated and analyzed during this study are included in this manuscript. Raw qualitative data are not available and will not be publicly shared, as this would compromise the anonymity and protection of our study participants.

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
