# Peer review of "Meaningful Engagement of Persons Affected by Leprosy in Research: An Exploration of Its Interpretation, Barriers, and Opportunities"

_tropicalmed, 2023, doi:10.3390/tropicalmed8010052_

Round 1

Reviewer 1 Report

The manuscript of de Groot et al. is an interesting qualitative study of leprosy focused on the experience of leprosy patients.

The study is well-written and has interesting issues.

My suggestion to improve this study is to include a more detailed explanation of these types of studies and their relevance in the Discussion and the Introduction, considering that most readers of this journal need to understand these types of studies and their differences with quantitative analysis appropriately.

Also, Figure 3 has words that surpass other words, which limits the proper understanding (I wonder if this is caused because the original figure has these alterations or because these alterations were caused due to changes in the pdf file).

Also, check the format of the references. Some of them need to be in the format required for this journal.

After these modifications, this article could be accepted for publication in the TropicalMed journal.

Reviewer 2 Report

This survey report is descriptive and well-constructed emphasizing the problem faced by leprosy patients. This review is different from the data collected from the patients as it involves scientific personnel, especially those dedicated to leprosy research. The author also considered their funding status which further helped to understand how actively they are involved in the field. The survey data in a similar manner need to be collected more often to involve patients and to include their perspectives while designing new therapies and treatments.

I have the following minor comments about the study:

  •  This article is more appropriate for a survey report category rather than a research article. Because it includes the researcher's view and their experiences related to Leprosy. While keeping it under the research article category would confuse the readers. In another way, a scientific professional working on leprosy would have already published his data in a research article.
  • In line 148 author mentioned that participants got the opportunity to look into the data and do the modification. This process could have led to the data business. 
  • Line 163 suggested that the survey question improved while doing the survey. In this case, that data was recollected from the personnel those already participated in the study.
  • In line 174 author mentioned that "The origin of the LRI-funded researchers who refused to participate in the study were not systematically different from those who participated". What was the basis for this interpretation? 
  • While talking about the severe impact of leprosy, there is still available treatment that requires discussion while providing constructive suggestions for improvement. Here the problem is addressed in sufficient detail however in-depth discussion to deal with the problem would be more appreciated.
  • It would be great to include that section to spread awareness among patients.
  • Along with this adopting a precautionary mindset (hygiene and cleanness) could significantly decrease the chances of infection and avoids disease severity. The awareness among the patients could be pivotal in breaking the tabu associated with Leprosy.
  • One limitation of this survey was that it did not include patients from the African continent and South East Asia regions where the literacy rate is low and the Middle Eastern regions where women are still fighting for their rights. The inclusion of patients along with scientific personnel could have significantly improved this study. 
  • Please see the attached file for further comments on grammatical and typographic errors.

Round 2

Reviewer 2 Report

Please check the graph legend for Figure 3, as some overlapping is there in the text.

The authors have rectified the issue and justified the comments raised in the previous version.